# Polyamine detergents tailored for native mass spectrometry studies of membrane proteins

Yun Zhu[1,2], Bo-Ji Peng [1,2], Smriti Kumar[1,2], Lauren Stover [1,2], Jing-Yuan Chang[1], Jixing Lyu[1], Tianqi Zhang[1], Samantha Schrecke[1], Djavdat Azizov[1], David H. Russell [1], Lei Fang [1] ✉ & Arthur Laganowsky [1] ✉

Native mass spectrometry (MS) is a powerful technique for interrogating membrane protein complexes and their interactions with other molecules. A key aspect of the technique is the ability to preserve native-like structures and noncovalent interactions, which can be challenging depending on the choice of detergent. Different strategies have been employed to reduce charge on protein complexes to minimize activation and preserve non-covalent interactions. Here, we report the synthesis of a class of polyamine detergents tailored for native MS studies of membrane proteins. These detergents, a series of spermine covalently attached to various alkyl tails, are exceptional charge-reducing molecules, exhibiting a ten-fold enhanced potency over spermine. Addition of polyamine detergents to proteins solubilized in maltoside detergents results in improved, charge-reduced native mass spectra and reduced dissociation of subunits. Polyamine detergents open new opportunities to investigate membrane proteins in different detergent environments that have thwarted previous native MS studies.

Native mass spectrometry (MS) of membrane protein complexes over the past decade has provided insight into stoichiometry, ligand and drug binding, and protein stability. Membrane proteins are typically purified in detergent to maintain their solubility. During MS analysis, the protein-detergent complexes are transferred from the solution phase to the gas phase through nanoelectrospray ionization (nESI)[1]. Membrane proteins are then stripped of detergent through collisional activation, in which the amount of activation energy required to release proteins is dependent on the detergent[2,3]. Previous studies on protein ejection and subunit dissociation suggest that the dissociation energy is directly correlated with the amount of charge on the complex[4–7]. The dissociation of detergents is thought to serve as an alternative pathway to relieve the Coulombic repulsion instead of protein unfolding/dissociation[7,8]. Under suitable conditions, non-covalent interactions such as ligand-bindings can be preserved such that various biophysical parameters (e.g. equilibrium dissociation constants) can be measured[9–13].

Protein complexes tend to dissociate and/or unfold as the charge density increases to counter the increasing intermolecular Coulombic repulsion[3–8,14]. Instrument settings required to eject membrane proteins from non-charge-reducing detergents, such as the commonly used dodecyl maltoside (DDM), often leads to dissociation of subunits and unfolding of protein complexes[2,3]. This result has created a barrier to native MS studies of membrane proteins in these detergent environments. Therefore, recent effort has been spent on the discovery and development of charge-reducing detergents[15–23] to help preserve non-covalent interactions and native-like structures. Detergents such as C8E4 (tetraethylene glycol monooctyl ether) and LDAO (Lauryldimethylamine oxide) exhibit charge-reducing properties in the nESI process and are often released from membrane proteins at a much lower collisional activation energy compared to non-charge-reducing detergents[3]. The synthesis of oligoglycerol detergents containing various basic groups has recently been reported[19,22–24]. These

[1]Department of Chemistry, Texas A&M University, College Station, TX 77843, USA. [2]These authors contributed equally: Yun Zhu, Bo-Ji Peng, Smriti Kumar, Lauren Stover. ✉e-mail: fang@chem.tamu.edu; ALaganowsky@chem.tamu.edu

detergents displayed modest charge-reducing properties and the reduction in charge was less than that observed for some commercially available detergents, such as C8E4[19,22–24]. Another approach involves the addition of charge-reducing molecules such as trimethylamine N-oxide (TMAO) and polyamines[13,18,20]. These molecules appear to be surface-active in the ionization process where they carry protons away from the droplet, thereby reducing charge on proteins[16,18]. A downside to using the charge-reducing molecules is that, even for the most potent molecules (e.g., spermine), high concentrations are required to achieve a desirable result[18]. Another drawback is the limited compatibility of charge-reducing additives with different detergents[18,20].

We envisioned that a detergent containing a polyamine head group and hydrophobic alkyl tail can serve the purpose of membrane protein solubilization and charge-reducing simultaneously. Herein, we report the synthesis of a series of polyamine detergents derived from spermine (SPM). A subset of membrane proteins that vary in topology, stoichiometry, and number of transmembrane helices were used to determine if the tailored detergent displays ideal properties for native MS studies.

## Results and discussion

### Rationale and design of detergents tailored for native MS

Our first objective was the design and chemical synthesis of detergents tailored for native MS studies. From our view, the desirable attributes of a detergent for native MS include (i) charge-reducing properties and (ii) easy release from the target membrane protein. To this end, we aimed to covalently link the potent charge-reducing polyamine, spermine, with various hydrocarbon chains [linear *n*-octyl (C8), *n*-decyl (C10), *n*-dodecyl (C12) chain, or double *n*-butyl (C4)$_2$ chains] to balance the ease of release. In this context, four detergent compounds C8SPM, C10SPM, C12SPM, and (C4)$_2$SPM were designed and synthesized (Fig. 1a). For example, the synthesis of C10SPM (Fig. 1b) started with the alkylation of spermine derivative **1**, in which one of the primary amino group is free while all other amino groups are protected by tert-Butyloxycarbonyl (Boc) groups[25]. Next, covalent coupling of **1** with decyl alkyl group was conducted. Previously reported synthesis of similar compounds relied on direct alkylation of primary amine with bromoalkane[26]. However, potential over-alkylation of the amine group into tertiary or even quaternary nitrogen center could be a challenge for

obtaining pure product in a reasonable yield. Here, reductive amination was employed to couple spermine and the alkyl tail in a controllable manner to limit over-alkylation. Imine-condensation between compound **1** and decanal **2**, followed by in-situ reduction by sodium cyanoborohydride, afforded Boc-C10SPM in 93% yield. Subsequently, treatment of Boc-C10SPM by excessive trifluoroacetic acid (TFA) led to deprotection of all the Boc groups, affording the polyamine detergent product C10SPM as a TFA salt. To purify the product, a reverse-phase flash column chromatography was applied with gradient eluent of 2% acetic acid in water/methanol mixture to afford C10SPM with high purity in 57% yield. C8SPM, C12SPM, and (C4)$_2$SPM were synthesized and purified using a similar procedure (Supplementary Fig. 1). The overall yields after purification range from 15% to 44% (see Supplementary Note 1).

### SPM-derived detergents as charge-reducing additives

To examine the charge-reducing potency of these SPM-derived detergents as additives, two different membrane proteins were titrated with SPM and SPM-derived detergents. The native mass spectrum of TRAAK (TWIK-Related Arachidonic Acid-stimulated K$^+$ channel), a dimeric mammalian two-pore domain potassium channel[13,27], solubilized in the charge-reducing detergent C10E5 (pentaethylene glycol monodecyl ether) had a weighted average charge state ($Z_{avg}$) of $13.92 \pm 0.16$ (Fig. 2, Supplementary Fig. 2, and Supplementary Table 2). Consistent with previous studies[13,27], the addition of 10 mM SPM reduced the overall charge ($Z_{avg}$) of TRAAK to $9.82 \pm 1.45$. The addition of SPM and all SPM-derived detergents to TRAAK showed comparable charge reduction effect at low additive concentrations. However, at high additive concentrations (>1 mM), the detergents with longer alkyl chains (C10SPM and C12SPM) showed enhanced charge reduction, with the lowest $Z_{avg}$ of $7.82 \pm 0.53$ achieved by adding 10 mM C10SPM as the additive (Fig. 2c). Analogous experiments revealed similar results for Aquaporin-Z (AqpZ), a tetrameric, bacterial water channel, and solubilized in the charge-reducing detergent C8E4[28,29]. A more pronounced charge-reducing effect was observed for AqpZ with C10SPM and C12SPM at higher concentrations (Fig. 2d, Supplementary Fig. 3, and Supplementary Table 3). Ion mobility measurements also show retention of compact conformers for charge-reduced ions for both SPM and SPM-derived detergents (Supplementary Fig. 4). These results indicate improvements in charge-reducing properties of SPM-

**Fig. 1 | SPM detergents and their synthesis. a** Structures of spermine (SPM) and SPM-derived detergents C8SPM, C10SPM, C12SPM, and (C4)$_2$SPM. **b** Synthesis of C10SPM.

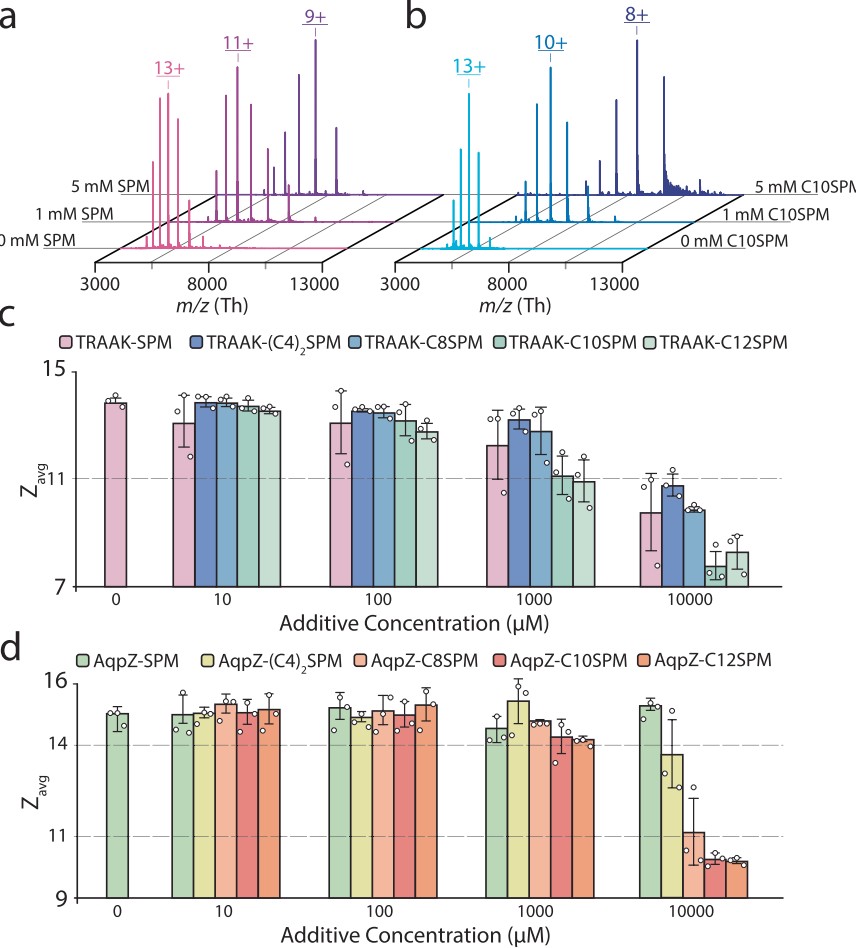

**Fig. 2 | SPM detergents display enhanced charge-reducing properties compared to SPM. a**, **b** Example native mass spectra of TRAAK in C10E5 with the addition of (**a**) SPM and (**b**) C10SPM. **c**, **d** Bar graph of the weighted average charge states ($Z_{avg}$) for (**c**) TRAAK and (**d**) AqpZ in the presence of different concentrations of SPM, (C4)$_2$SPM, C8SPM, C10SPM, and C12SPM. Reported are the mean and standard deviation ($n = 3$, biological replicates). Source data are provided as a Source Data file.

derived detergents with longer alkyl chains compared to that of those with shorter chain(s).

## Membrane proteins solubilized in SPM detergents

Motivated by the excellent performance of SPM-containing detergents, we investigated the charge-reducing properties of membrane proteins solubilized in C10SPM alone, and C10SPM mixed with other charge-reducing detergents (Fig. 3 and Supplementary Fig. 5). The mass spectrum of TRAAK solubilized in C10SPM showed a significant reduction in charge ($Z_{avg} = 7.8 \pm 0.5$) compared to that in C10E5 (Fig. 3b and Supplementary Table 4). Notably, when C10SPM was used as an additive, it required nearly six-fold higher concentration to achieve a similar $Z_{avg}$, indicating the detergents compete to solubilize the membrane protein. The charge-reducing effect of C10E5 and C10SPM mixture at various concentrations was also investigated. Compared to TRAAK in C10E5, the mixture containing 0.8 mM C10E5 and 0.9 mM C10SPM displayed a decrease of $3.2 \pm 0.5$ in $Z_{avg}$, which was comparable to that of C10SPM as an additive at a similar concentration ($\Delta Z_{avg} = 2.1 \pm 0.74$; Figs. 2 and 3, and Supplementary Table 2–4). As the fraction of C10SPM increases, a gradual decrease in $Z_{avg}$ was observed (Fig. 3b). Similar results were observed for AqpZ in C10SPM and C8E4 mixtures (Fig. 3c, d). Interestingly, for TRAAK, equal and higher concentrations of C10SPM ($\geq 0.9$ mM) resulted in a similar $Z_{avg}$ value (Fig. 3b). However, for the mixed detergent environment a bimodal charge distribution is observed. This result is consistent for conditions that contain molecules with different charge-reducing properties, such

as a mixture of DDM and C8E4 or soluble proteins in mixed volatile buffers[3,16]. For increasing ratios of C10SPM to C8E4, a continuous reduction in charge was observed for AqpZ (Fig. 3d). The variation in the charge-reduction of C10SPM in mixed detergent environments could stem from the ten-fold higher critical micelle concentration (CMC) of C8E4 compared to C10E5 (Supplementary Table 1). Taken together, C10SPM can solubilize membrane protein complexes and exhibit improved charge-reducing properties, especially in mixed micelles.

## Compatibility of SPM detergents with commonly used detergents

Maltoside detergents represent some of the most frequently used detergents in membrane protein research. However, native MS of proteins solubilized in these detergents often leads to dissociation of subunits/ligands and unfolding. For example, the native mass spectrum of AqpZ in decyl maltoside (DM) was dominated by signals corresponding to dissociated species (monomers and dimers) (Fig. 4a). However, the addition of 1 mM SPM resulted in increased population of intact, tetrameric AqpZ. A mixture of 1x CMC DM and 1 mM C10SPM produced tetramers and a small contribution for monomers (Fig. 4a). Similar observations were made for the other SPM-derived detergents (Fig. 4a and Supplementary Fig. 6a). The native mass spectrum of TRAAK in DM showed signals for highly charged homodimers (Fig. 4b). The absence of dissociated species is anticipated as TRAAK contains a conserved cysteine in the helical cap that forms a disulfide,

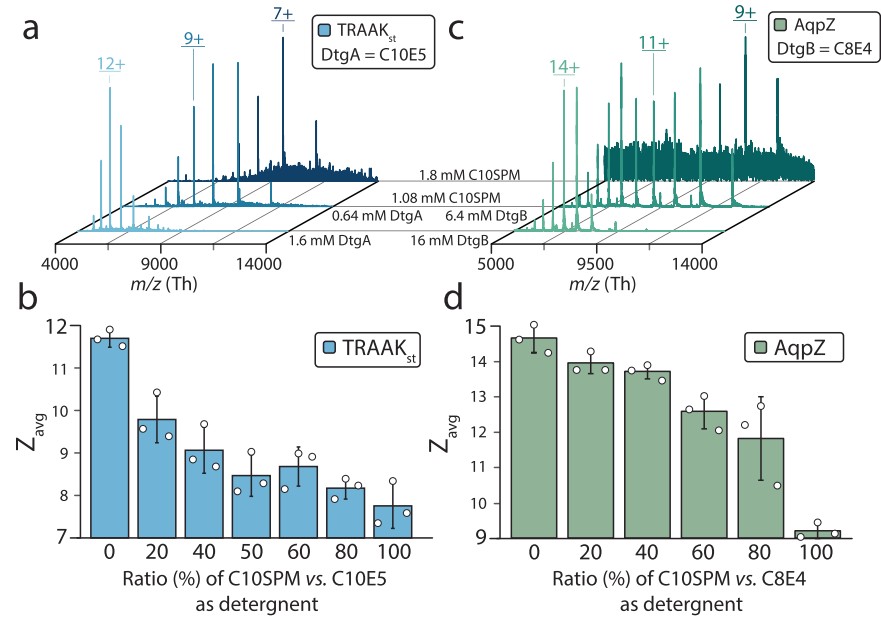

**Fig. 3 | Mixed micelles containing C10SPM display varied charge reduction.** Native mass spectra of (**a**) TRAAK$_{st}$ with various ratios of C10E5/C10SPM as detergent, and (**c**) AqpZ with various ratios of C8E4/C10SPM as detergent. 100% ratio for C10E5 (detergent A) is 1.6 mM, 100% ratio for C8E4 (detergent B) is 16 mM, 100% ratio for C10SPM is 1.8 mM. Weighted average charge ($Z_{avg}$) for each detergent composition for (**b**) TRAAK$_{st}$ in C10E5/C10SPM and (**d**) AqpZ in C8E4/C10SPM are summarized in bar graphs. Reported are the mean and standard deviation ($n = 3$, biological replicates). Source data are provided as a Source Data file.

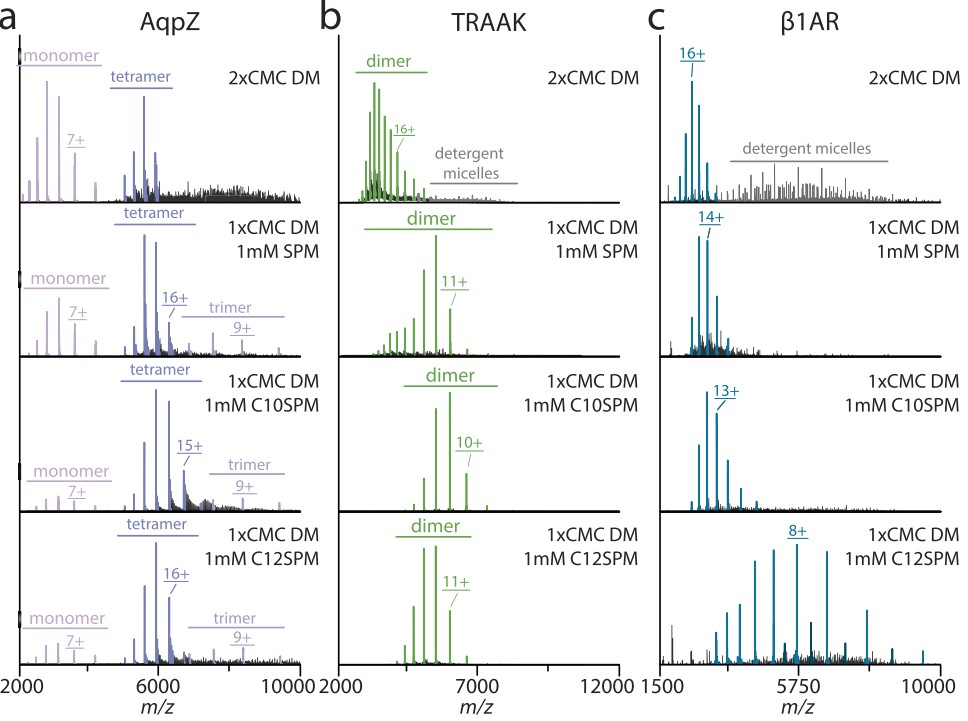

**Fig. 4 | Charge-reducing effect on different proteins in DM with the addition of SPM and various SPM detergents. a** Mass spectrum of AqpZ under different conditions. Going from top to bottom are spectra collected with: 2xCMC DM, 1xCMC DM with 1 mM SPM, 1xCMC DM with 1 mM C10SPM, 1xCMC DM with 1 mM C12SPM. **b** Mass spectrum of TRAAK under different conditions. Shown as described for panel **a**. **c** Mass spectrum of β1AR under different conditions. Shown as described for panel **a**.

crosslinking the subunits[30]. The addition of 1 mM SPM to TRAAK in DM reduced the $Z_{avg}$ for the dimer by more than two-fold (Fig. 4b). Charge reduction was also achieved for TRAAK in a mixture of 1x CMC DM and 1 mM C10SPM, and comparable to that doped with 10 mM SPM (Fig. 4b and Supplementary Fig. 6b). Notably, TRAAK in DM mixed with an

equal part of C10E5, a commonly used charge-reducing detergent, only showed a slight charge reducing effect (Supplementary Fig. 7). A thermostabilized version of the turkey beta-1 adrenergic receptor (β1AR), a monomeric G-protein-coupled receptor (GPCR)[31–36], purified in DM had a charge state distribution centered around 16 (Fig. 4c and

Supplementary Fig. 6c). The addition of 1 mM SPM had no effect whereas 10 mM SPM resulted in a significant charge reduction (Supplementary Fig. 6c). β1AR examined in the same mixture of DM and C10SPM as described above did not show significant charge reduction (Fig. 4c). For all three membrane proteins, the performance of C10SPM and C12SPM were comparable. An exception was β1AR where C12SPM resulted in a dramatic shift in $Z_{avg}$. These results reveal that SPM-derived detergents containing C10 and C12 tails outperformed SPM and, in some cases, required 10-fold less of the polyamine detergent to achieve a comparable result.

Our previous work showed the incompatibility between SPM and DDM[18], wherein the mixture produced an unresolved mass spectrum of the membrane protein. The bacterial trimeric ammonia channel (AmtB) in DDM required significant collisional activation to obtain an interpretable mass spectrum. Under these harsh conditions, signals for dissociated monomers dominated along with some intact trimers (Fig. 5a). The addition of 1 mM SPM to AmtB in DDM resulted in an uninterpretable hump (Fig. 5a). However, AmtB solubilized in 1x CMC DDM and 1 mM C10SPM showed a significant depletion of the signal for the dissociated monomer (Fig. 5a). Comparable results were observed for the other SPM-derived detergents (Supplementary Fig. 8a). Like AmtB, the mass spectrum of the TWIK-related acid sensing potassium channel (TASK1) in DDM corresponded to dissociated monomers (Fig. 5b). When mixed with 1 mM C10SPM, the monomer signals were diminished, and a lower charge state distribution for the homodimer was observed (Fig. 5b). This result was not obtained using SPM in place of C10SPM (Fig. 5b and Supplementary Fig. 8b). The mass spectrum of AqpZ in DDM had abundant signals for dissociated monomers, and the spectral peaks for the tetrameric channel were poorly resolved (Fig. 5c). The addition of 1 mM SPM to AqpZ produced uninterpretable mass spectrum. In contrast, AqpZ in a mixture of DDM and 1 mM C10SPM (or other SPM-derived detergent) produced a well-resolved mass spectrum (Fig. 5c). Interestingly, the monomeric protein β1AR in DDM had a $Z_{avg}$ that was lower than that for the protein in DM

(Figs. 4c and 5d). Mass spectra of β1AR in DDM and in the presence of SPM and SPM-derived detergents resulted in no significant reduction in charge states (Fig. 5d and Supplementary Fig. 8d). Taken together, these results demonstrate the utility of SPM-derived detergents to charge reduce membrane proteins solubilized in DDM.

This work used a variety of membrane proteins, from monomers to tetramers, in different environments to provide insight into conditions suitable for native MS studies. The addition of SPM, and SPM-derived detergents, charge-reduced membrane proteins solubilized in C8E4 and C10E5. Membrane proteins solubilized in C10SPM and C12SPM showed the lowest charge states, demonstrating the enhanced potency of these polyamine detergents. For example, to achieve a similar charge state distribution of TRAAK in C10SPM would require a 6-fold higher concentration of SPM. The commonly used DM and DDM detergents have limited applications in native MS studies due to high charge densities, which readily leads to dissociation of protein complexes. The addition of both SPM and SPM-derived detergents to proteins in DM resulted in resolved, charge-reduced mass spectra. However, membrane proteins in DDM supplemented with SPM resulted in uninterpretable humps. This suggests a competition between protein-detergent complexes and charge-reducing molecules. Unlike SPM, the addition of SPM-derived detergents to membrane proteins solubilized in DDM resulted in well-resolved, charge-reduced mass spectra. Despite DM having a 10-fold higher CMC than DDM, the results suggest that DM forms weaker interactions with membrane proteins, enabling SPM and SPM-derived detergents to engage with the protein to form charge-reduced ions. However, DDM-protein interactions are stronger than DM, and the micellar belt shields interactions with SPM. The alkyl chain of SPM-derived detergents enables the molecule to interact with the membrane protein.

In summary, among SPM and the tailored SPM-derived detergents, the ones modified with longer alkyl chains (C10SPM and C12SPM) are more potent charge-reducing molecules for membrane proteins. The mixed micelles containing C10SPM and C12SPM showed

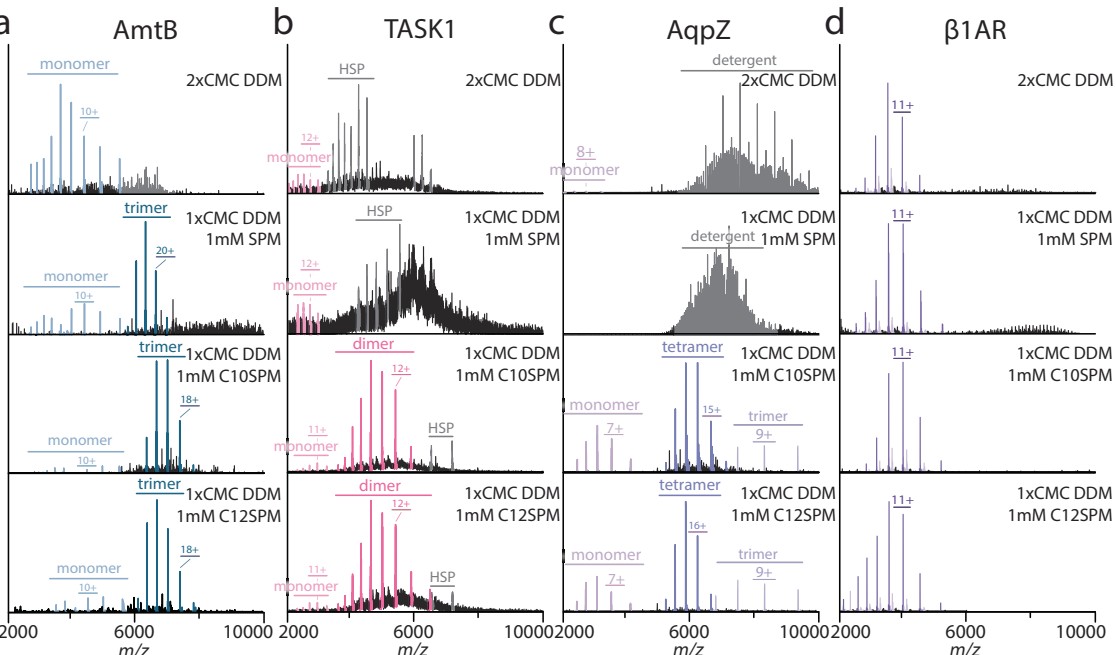

**Fig. 5 | Charge-reducing effect on different proteins in DDM with the addition of SPM and SPM detergents. a** Mass spectrum of AmtB in DDM is dominated by signal for dissociated monomers. The addition of SPM produced an unresolved mass spectrum whereas SPM detergents produced resolved signal for the trimeric channel. From top to bottom are spectra collected with 2xCMC DDM, 1xCMC DDM with 1 mM SPM, 1xCMC DDM with 1 mM C10SPM, 1xCMC DDM with 1 mM C12SPM. **b** Mass spectrum of TASK1 in DDM and with the addition of SPM and SPM-derived detergents. **c** Mass spectrum of AqpZ in DDM and in the presence of SPM and SPRM detergents. **d** Mass spectrum of β1AR in DDM and with SPM and in the presence of SPM or SPM detergents. Shown as described for **a**.

significantly improved mass spectral signals for intact membrane protein complexes, especially for membrane proteins in the non-charge-reducing maltoside detergents. As many membrane proteins are sensitive to the choice of detergents, the ability to generate charge-reduced ions with SPM-derived detergents in different detergent environments opens new and exciting possiblities.

## Methods

### Synthesis and characterization of polyamine detergents

All reagents were purchased from commercial suppliers and used as received unless otherwise stated. Trifluoroacetic acid was dried by addition of 0.01 eq. trifluoroacetic anhydride and let stir for 10 min at room temperature prior to use. $^1$H and $^{13}$C nuclear magnetic resonance (NMR) spectra were recorded on Bruker Avance 400/500 MHz spectrometers at room temperature and processed by MestReNova 14.2.2. Chemical shifts were reported in ppm relative to the signals corresponding to the residual non-deuterated solvents (for $^1$H NMR: CDCl$_3$ $\delta$ = 7.26 ppm; D$_2$O $\delta$ = 4.79 ppm; for $^{13}$C NMR: CDCl$_3$ $\delta$ = 77.16 ppm). For $^{13}$C NMR spectra recorded in D$_2$O, trifluoroacetate was used as an internal standard ($\delta$ = 164.88, 164.60, 164.32, 164.04). Normal phase flash column chromatography was carried out by using SiO$_2$ as the stationary phase. Reverse-phase flash column chromatography purifications were carried out using a Yamazen Science Inc Smart Flash W-Prep 2XY with C18-silica (ODS) cartridges. Electrospray ionization mass spectrometry (ESI-MS) experiments were performed using a Thermo Scientific Q-Exactive Focus operated in full MS in positive mode.

### Determination of the critical micelle concentration

The CMC values of the detergents were measured by adapting a literature-reported method[37]. An aqueous solution of pyrene (the reporter dye) was prepared by adding 1 µL of the pyrene solution (10 mM in THF) to 5 mL of ammonium acetate buffer (200 mM in water, pH = 7.4), affording a 2 µM aqueous pyrene solution. Multiple samples with variable detergent concentrations (0.88–58 mM) and fixed pyrene concentration (2 µM) were prepared from this solution. The samples were ultrasonicated for 5 min and then transferred to a cuvette for measurements of fluorescence spectra. For each detergent, the intensity ratio (I$_1$:I$_3$) of transition-forbidden band at 371 nm (I$_1$) to transition-allowed band at 383 nm (I$_3$) as a function of logarithm detergent concentration was plotted. The ratio goes from ~2.0 (below CMC) to ~1.2 (above CMC). The inflection point of the sigmoidal curve resulting from a Boltzmann fit of the data equals the logarithmic CMC value.

### Plasmid construction and protein expression for AmtB and AqpZ

AmtB from *E. coli* was expressed with an N-terminal HRV3C protease cleavable 10x His tag followed by maltose binding protein (MBP) from pCDFDuet-1 (Novagen) in *E. coli* C43 (DE3) (Lucigen) as previously described[31]. Cells were inoculated in terrific broth (TB, IBI Scientific) and the expression was induced with a final concentration of 0.5 mM IPTG when an O.D. of 0.6–0.8 was reached and allowed to grow for 18 h at 20 °C while shaking. AqpZ from E. coli was expressed with a C-terminal Strep-tag II from pET15B (Novagen) in BL21-A1 (Invitrogen). Cells were inoculated in TB and protein expression was induced with 0.2% arabinose at an O.D. of 0.6–0.8. The induced cells grew for 3 h at 37 °C while shaking then harvested by pelleting at 5000 × *g* for 10 min.

### Plasmid construction and protein expression for TRAAK

The expression and purification of TRAAK were similar as previously reported. In brief[13], TRAAK (K$_{2P}$4.1b) was transformed into *SuperMan5* expression strain (BioGrammatics Inc.) by electroporation following manufacturers protocol and high-level expressing transformants were selected via an adapted high-throughput expression screen. The

selected transformants were grown in 50 mL YPD overnight at 30 °C while shaking. The overnight culture was transferred to 600 mL of BMGY and was allowed to grow at 30 °C for 24 hrs while shaking. The BMGY cultures were spun down and media exchanged into BMMY, and was allowed to grow at 30 °C for 48 h. The cells were harvested and washed with K-lysis buffer (200 mM KCl, 50 mM TRIS, pH = 7.4 at room temperature) by centrifugation (2000 × *g* for 5 min).

### Plasmid construction and protein expression for TRAAK$_{st}$, TASK1, and β1AR

TRAAK$_{st}$, TASK1, and β1AR were expressed in insect cells. The expression plasmid for human TRAAK containing a C-terminal Strep-tag II has been previously reported[27]. The cDNA for human TASK1 (Uniprot O14649, residues 1-259) was purchased from DNASU Plasmid Repository (clone ID HsCD00871609). The coding region for residues 1-259 of TASK1 was subcloned into a modified pACEbac1 (Geneva Biotech) plasmid that introduces a C-terminal Strep-tag II. The coding region for β1AR (Uniprot P07700, residues 44-244, 272-366) with seven thermostabilizing mutations (R68S, C116L, E130N, D322K, F327A, F338M, and C358A) was codon optimized for insect cells and synthesized as a gBlock gene fragment (Integrated DNA Technologies). The gene was subcloned into a modified pACEbac1 (Geneva Biotech) insect cell expression plasmid with a C-terminal Strep-tag II. The transfection and infection of the insect cells were performed as previously described[27]. In brief, plasmids were transformed into DH10EMBacY cells (Geneva Biotech) using the manufacturer-recommended transformation protocol. The recombinant bacmids were purified using the HiPure Plasmid Midiprep kit (Invitrogen). *Spodoptera frugiperda* (Sf9) (Expression Systems LLC.) cells in suspension were transfected using PEI Max (Polysciences) transfection reagent. Protein expression was carried out by infecting *Trichoplusia ni* (Tni) (Expression Systems LLC.) cells with the recombinant virus and grown for 72 h at 27 °C while shaking. Insect cells were harvested at 4000 × *g* for 10 min.

### Protein expression and purification

**AmtB and AqpZ.** Cell pellets were resuspended in TBS buffer (150 mM NaCl, 50 mM Tris, pH 7.4 at room temperature) and were lysed using a Microfluidics M-110P microfluidizer operating at 25,000 PSI. The cell debris was pelleted through centrifugation for 25 min at 20,000 × *g* at 4 °C. The membranes were pelleted through centrifugation for 2 h at 100,000 × *g* at 4 °C. Membranes were resuspended in resuspension buffer (100 mM NaCl, 20 mM Tris, 20% glycerol, 5 mM BME (2-mercaptoethanol), pH 7.4 at room temperature) and extracted overnight with 5% octyl glucoside (OG) at 4 °C. The extracted protein was clarified through centrifugation at 40,000 × *g* for 20 mins at 4 °C. The supernatant was filtered with a 0.45-µ syringe filter (Pall Corp.). For AmtB, the sample was loaded onto a HisTrap HP 5 mL column (Cytiva) equilibrated with NHA-DDM buffer (200 mM NaCl, 20 mM Tris, 20 mM imidazole, 10% glycerol, 0.025% DDM, pH 7.4 at room temperature) and eluted with NHB-DDM buffer (100 mM NaCl, 20 mM Tris, 500 mM imidazole, 10% glycerol, 0.025% DDM; pH 7.4 at room temperature). The eluted AmtB fusion protein was loaded onto a 5 mL MBPTrap HP column (Cytiva) pre-equilibrated with the MBP-loading buffer (100 mM NaCl, 20 mM Tris, 10% glycerol, 0.025% DDM, pH 7.4 at room temperature) and eluted with MBP-elution buffer (100 mM NaCl, 20 mM Tris, 10 mM maltose, 10% glycerol, 0.025% DDM, pH 7.4 at room temperature). HRV3C protease was added to protein in a 100:1 protein-to-protease ratio, and the mixture was incubated overnight at 4 °C. It was loaded onto a HisTrap HP column equilibrated in NHA-DDM buffer, and the flow-through containing the tag-less protein was collected. The peak containing AmtB was pooled, aliquoted into 50 µL shots, flash-frozen in liquid nitrogen, and stored at −80 °C. For AqpZ, the extracted proteins were loaded onto a StrepTrap HP 5 mL column (Cytiva) equilibrated with Buffer-A (100 mM NaCl, 20 mM Tris, 10%

glycerol, 0.2% DM; pH 7.4 at room temperature) and eluted with Buffer-B (100 mM NaCl, 20 mM Tris, 2.5 mM desthiobiotin, 10% glycerol, 0.2% DM, pH 7.4 at room temperature). It was then concentrated using a 50 kDa MWCO concentrator (MilliporeSigma). For AmtB and AqpZ, concentrated protein was loaded onto a Superdex 200 Increase 10/300 GL column (GE Healthcare) equilibrated in GF buffer (100 mM NaCl, 20 mM Tris, 10% glycerol, 0.5% C8E4, pH 7.4 at room temperature).

**TRAAK.** The purification of TRAAK ($K_{2P}$4.1b) was same as previous study[13]. In brief, harvested cells were resuspended and lyse in K-lysis buffer with Microfluidics M-110P microfluidizer operating at 30,000 PSI. The cell debris was removed by centrifugation ($20,000 \times g$ for 20 mins). The membrane was formed by ultra centrifugation ($100,000 \times g$ for 2 h at 4 °C) and was resuspended and extracted in K-lysis buffer with 1% DDM overnight. The non-subitizable particles were removed by centrifugation ($20,000 \times g$ for 20 min at 4 °C) and syringe filtration (0.45-μ, Whatman). The filtered supernatant was purified with a HisTrap HP column equilibrated in KHA-DM (200 mM KCl, 50 mM Tris pH 7.8 at room temperature, 10% glycerol, and 0.2% n-decyl-β-D-maltopyranoside (DM)) and gradient elution was applied with KHB-DM (200 mM KCl, 50 mM Tris pH 7.8 at room temperature, 500 mM Imidazole, 10% glycerol, 0.2% DM). The his-elute was loaded onto the StrepTrap HP column pre-equilibrated in KHA-DM and was eluted with KHA-DM supplied with 3 mM desthiobiotin. The strep-elute was desalted into KHA-DM using a HiPrep 26/10 desalting column (GE health). Desalted samples were treated with TEV overnight at room temperature and purified with reverse IMAC in KHA-DM. The flowthrough was further purified, and buffer exchanged into KHA-C10E5 (200 mM KCl, 20 mM TRIS, 10% glycerol 0.062% pentaethylene Glycol Monodecyl Ether (C10E5), pH 7.8 at room temperature) with a Superdex 200 Increase column (GE). The sample was pooled and analyzed by native mass spectrometry.

**TRAAK$_{st}$ and TASK1.** The purification protocol for TRAAK$_{st}$ has been previously published[27]. In brief, cell pellets were resuspended in K-lysis and were lysed by passing through a microfluidizer. Cell debris was removed by centrifugation at $25,000 \times g$ for 20 min, and the supernatant was centrifuged again at $100,000 \times g$ for 2 h to pellet membranes. The membrane was resuspended in lysis buffer and was extracted with 1% DDM (w/v) for 2 hrs at 4 °C while rotating. The extracted protein was clarified by centrifugation at $20,000 \times g$ for 5 min, and the supernatant was filtered using a 0.45-μ syringe filter (Whatman). The clarified protein was loaded onto a StrepTrap HP column (Cytiva) pre-equilibrated in KHA-DM (200 mM KCl, 50 mL Tris, 10% v/v glycerol, 0.2% w/v DM, pH = 7.4 at room temperature) and was eluted with KHA-DM supplied with 3 mM desthiobiotin. Eluted protein was desalted with a HiPrep 26/10 desalting column (GE health) in DM-desalting buffer (150 mM KCl, 20 mM Tris, 10% v/v glycerol, 0.2% w/v DM, pH = 7.4 at room temperature). To prepare TRAAK in C10E5, after loading onto the StrepTrap HP column, the protein was washed with KHA-DM followed by KHA-C10E5 (200 mM KCl, 50 mL Tris, 10% v/v glycerol, 0.062% w/v C10E5, pH = 7.4 at room temperature) and eluted with KHA-C10E5 supplemented with 3 mM desthiobiotin. The desalting was carried out in C10E5-desalting buffer (150 mM KCl, 20 mM Tris, 10% v/v glycerol, 0.062% w/v C10E5, pH = 7.4 at room temperature). Proteins were aliquoted and stored at −80 °C. TASK1 was purified as described for TRAAK$_{st}$.

**β1AR.** Cell pellets were resuspended in TBS with a protease inhibitor cocktail tablet (cOmplete, EDTA-free) dissolved. Resuspended cells were lysed, and cell debris was removed by centrifugation. Membranes were pelleted from the supernatant, resuspended in TBS, and extracted with 2% DDM for 2 h at 4 °C while rotating. Extracted protein was clarified by centrifugation, and the supernatant was filtered. Proteins

were purified at 4 °C with a StrepTrap HP column pre-equilibrated in NHA-DDM (350 mM NaCl, 20 mM TRIS, 1 mM EDTA, 0.025% w/v DDM, pH = 7.4 at room temperature), eluted with NHA-DDM supplemented with 3 mM desthiobiotin, and desalted with a HiTrap 5 mL desalting column in DDM-Na-desalting buffer (100 mM NaCl, 20 mM TRIS, 0.2 mM EDTA, 0.025% w/v DDM, pH = 7.4 at room temperature). Proteins were aliquoted and stored at −80 °C.

### Sample preparation for MS analysis
MS samples were prepared as previously described[10]. In brief, proteins were buffer exchanged into MS buffer (200 mM ammonium acetate, pH = 7.4) supplemented with 2xCMC detergent using a centrifugal desalting column (Micro Bio-Spin 6 Columns, Bio-Rad). The concentration of proteins was determined using DC protein assay (Bio-Rad) with BSA as the protein standard.

### Native mass spectrometry
Native mass spectrometry was performed as previously described[13]. In brief, samples were loaded into a gold-coated borosilicate glass capillary and was introduced into an Exactive Plus Orbitrap mass spectrometer (Thermo Scientific) with Extended Mass Range (EMR) via nanoelectrospray ionization. Instrument settings were optimized for each sample and provided in Supplementary Table 5.

### Native ion mobility mass spectrometry
All AqpZ SPM and SPM-derived detergents titration data (Figs. 2 and 3) were collected on a SYNAPT G1 HDMS (Waters). The spectra for SPM and SPM-derived detergents titration were collected with following conditions: 1.7 kV capillary voltage, 150 for sampling cone, 10 for extraction cone, the source temperature was set to 90. Trap CE was set to 50 and Transfer CE was set to 20. The traveling wave parameters are as follows: TRAP DC entrance 5, bias 25, IMS DC entrance 10 and exit 9, Transfer DC entrance 5 and exit 9, source wave velocity 300 m/s and 10 V for the wave height, Trap wave velocity 300 m/s and 1.8 V for the wave height, IMS wave velocity 300 m/s and 18 V for the wave height, transfer wave velocity 100 m/s and 10 V for the wave height.

### SPM and SPM detergents titration
SPM was purchased from Cayman Chemical Company (Cat.: 18041). The stocks of SPM and SPM-derived detergents were prepared in Milli-Q water at desired concentrations, respectively. Serial dilutions of SPM and SPM-derived detergents were performed in MS buffer with corresponding detergents for the target protein (i.e., C10E5 for TRAAK, C8E4 for AqpZ). Proteins were mixed with SPM or SPM-derived detergents and incubated for 1–2 min at room temperature prior to the MS analysis.

### TRAAK$_{st}$ and AqpZ detergent exchange
TRAAK$_{st}$ was immobilized onto a drip column packed with Strep-Tactin Sepharose beads pre-equilibrated in the K-wash buffer (200 mM KCl, 20 mM Tris, pH = 7.4 at room temperature) supplemented with target detergents. Immobilized protein was washed with 10 CVs of wash buffer with target detergent and was eluted with 3 mM desthiobiotin in the same buffer. Eluted protein was buffer exchanged into MS buffer with the target detergent for further MS analysis. A similar process was performed for AqpZ, except the wash buffer was Na-wash buffer (100 mM NaCl, 20 mM Tris, pH = 7.4 at room temperature).

### Mixed micelles containing SPM-derived detergents
TRAAK$_{st}$ was detergent exchanged and buffer exchanged into both C10E5 and C10SPM in MS buffer. The concentration of TRAAK$_{st}$ in each detergent was determined by protein assay and was adjusted so that the protein concentrations were equal in both detergents. TRAAK$_{st}$ in C10E5 and C10SPM were mixed at different ratios and incubated for

1-2 min at room temperature prior to MS analysis. A similar procedure was used for AqpZ except C8E4 was used.

### SPM and detergent tail fused SPM with non-charge reducing detergents

Proteins were either purified (TRAAK, TASK1, AmtB, and β1AR) or detergent exchanged (AqpZ) into non-charge reducing detergent (DM or DDM). Each protein was prepared for native MS study as described previously in corresponding non-charge reducing detergent. For SPM-derived detergents spectra, the protein prepared in 2xCMC DM/DDM was mixed with an equal volume of 2 mM SPM-derived detergents MS buffer to make a final sample with 1xCMC DM/DDM and 1 mM SPM-derived detergents. For SPM addition, SPM stock was diluted with MS buffer with appropriate detergent at a concentration twice the final concentration. Protein was mixed with an equal volume of SPM. Mixed samples were incubated for 1–2 min at room temperature prior to MS analysis.

**Data analysis.** Raw spectra were processed with UniDec[38]. The weighted average charge ($Z_{avg}$) was computed using UniDec. The deconvolution parameters were set to no smoothing, m/z range 1500–12,000, charge range 5–30, mass sampling every 1 Da, and peak FWHM of 0.85.

## Data availability

Native MS data has been deposited at Zenodo (https://doi.org/10.5281/zenodo.8230111). A Source Data File is provided with this paper. Other data associated with this manuscript is available upon request.

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

## Acknowledgements
This work was supported by National Institutes of Health (NIH) under grant numbers (R01GM121751, R01GM138863, and RM1GM145416 to A.L.; R01GM139876 to A.L. and L.F.; and P41GM128577 to D.R.).

## Author contributions
Y.Z., B.P., L.F. and A.L. designed the research. B.P. and D.A. synthesized the charge reducing detergents and performed the CMC measurements. Y.Z. and S.S. optimized TRAAK and TRAAKst for mass spectrometry study. Y.Z. and J.C. expressed and purified TRAAK, TRAAKst, TASK1, and β1AR. Y.Z., J.L. and T.Z prepared the cell cultures for expression. L.S. expressed and purified AmtB. L.S. and J.C. expressed and purified AqpZ. Y.Z., J.C. and S.K. performed mass spectrometry experiments. Y.Z., B.P., L.F. and A.L. analyzed the data. Y.Z., B.P., L.F. and A.L. wrote the manuscript with input from the other authors.

## Competing interests
The authors declare no competing interests.
