## [Peer Review File · Nature Communications]

REVIEWERS' COMMENTS

Reviewer #1 (Remarks to the Author):

Native mass spectrometry assumes that proteins retain their native-like structure into the gas-phase. To achieve this, minimising the loss of native-like structures upon mass spectrometry analysis is key. The manuscript by Laganowsky and co-workers describes the synthesis of spermine-derived polyamine detergents that have been designed specifically to help maintain membrane protein structure upon mass spectrometry analysis. The introduction to the manuscript is clear and targeted well for a broad audience such as nature communications. The methodology is clear. The results are described well and will be of high interest to mass spectrometrists studying membrane proteins. Overall, the manuscript is of a very high standard and I recommend it to be published providing the two minor revisions below are addressed.

1) To strengthen the manuscript, it would be good to include how their new detergents may affect protein conformation in solution. Moreover, although unlikely, the effect that is observed could be due to the protein mis-folding within the membrane and thus why differences are observed in the mass spectra – more reference to the literature could help here.

2) The authors should also comment on why broad charge state distributions can be observed with their detergent added e.g. Figure 3C bottom panel. If the protein is more native, one might expect a more compact distribution of charges (i.e. 4-5 charge states rather than ~13). Also, if multiple conformations are present of the protein contributing to a varied charge state distribution, then why is only one state observed without the standard detergent (e.g. Figure S1E to and bottom spectra comparison)?

Reviewer #2 (Remarks to the Author):

The authors here report the synthesis and characterization of four different detergents for native mass spectrometry of membrane proteins. The detergents are composed of polyamine [spermine (SPM)] head group with different hydrophobic alkyl groups: (C4)2SPM, C8SPM, C10SPM and C12SPM. The four derivatives of SPM, along with SPM, are tested for their charge reducing potencies on a variety of membrane proteins that are expressed in cultured cells and purified. Proteins ranging from monomeric to tetrameric complexes, including TRAAK, AqpZ, β 1AR, AmtB, and TASK1 are tested.

C10SPM and C12SPM are found to be more potent charge-reducing detergents than SPM, C8SPM, and (C4)2SPM. The compatibilities of C10SPM and C12SPM with other commonly used detergents, including decyl maltoside and dodecyl maltoside, are tested and are found to be compatible. In addition, the manuscript also reports weighted average charge states for proteins, TRAAK and AqpZ, in the presence of different ratios of C10SPM to commercial detergents, C10E5 and C8E4, respectively. The mixed micelles display varied charge reduction depending on the ratio of the two detergents, while 100% of C10SPM (1.8 mM) reduces the average charge most (Figure 2). This is mostly true in the case of TRAAK, but the spectrum for AqpZ in 100% C10SPM (Figure S4), however, shows very broad features and makes it difficult for one to see the peaks at lower m/z, thus making it impossible to include those peaks in the calculation of the average charge state. The protein's compact conformers are retained to some extent as per IMS results (Figure S3) though.

From a brief comparison of this study with other studies on charge-reducing additives, listed in the references of this manuscript, it seems that the amount of additive (detergent) required in this study to observe a charge reduction is only 1 mM in most experiments, in comparison to the 40 mM (Imidazole Derivatives, <https://pubs.acs.org/doi/10.1021/acs.analchem.9b04263>), 100 mM (SPM and others, <https://pubs.acs.org/doi/10.1021/acs.analchem.0c01826>) and 100-250 mM (Trimethylamine oxide, <https://pubs.acs.org/doi/10.1007/s13361-019-02187-6>) in previous studies. At 10 mM concentration of various detergents tested here, the spectra of some proteins show broad features (Figures S1, S2, and S4). Notable charge reduction is observed in most cases, except for β 1AR in the presence of the detergent dodecyl maltoside (Figure 4D). The abundance of dissociated species in the protein spectra (Figures 3 and 4) in the presence of decyl maltoside or dodecyl maltoside is reduced

due to the addition of C10SPM or C12SPM, owing to their charge reducing properties.

This is not the first time I see this manuscript. The authors have added a significant amount of new data and with that converted the manuscript from a simple proof-of-concept into a truly systematic study. The detergents developed and tested here can simultaneously serve the purpose of membrane protein solubilization and charge reduction. This makes them a valuable addition to the native MS toolbox. I am therefore happy to recommend publication of the manuscript in Nature Communications.

Minor correction: Reference 26 is a duplicate of reference 16.

We thank the reviewers for their critical assessment of our manuscript and insightful feedback. We have responded to comments below.

Reviewer #1 (Remarks to the Author):

Native mass spectrometry assumes that proteins retain their native-like structure into the gas-phase. To achieve this, minimising the loss of native-like structures upon mass spectrometry analysis is key. The manuscript by Laganowsky and co-workers describes the synthesis of spermine-derived polyamine detergents that have been designed specifically to help maintain membrane protein structure upon mass spectrometry analysis. The introduction to the manuscript is clear and targeted well for a broad audience such as nature communications. The methodology is clear. The results are described well and will be of high interest to mass spectrometrists studying membrane proteins. Overall, the manuscript is of a very high standard, and I recommend it to be published providing the two minor revisions below are addressed.

1) To strengthen the manuscript, it would be good to include how their new detergents may affect protein conformation in solution. Moreover, although unlikely, the effect that is observed could be due to the protein mis-folding within the membrane and thus why differences are observed in the mass spectra – more reference to the literature could help here.

The average charge state of protein complexes is directly related to solvent accessible surface area. If the protein was unfolding, we would anticipate higher (not lower) charge states, which is not the case. The manuscript already cites several key papers.

2) The authors should also comment on why broad charge state distributions can be observed with their detergent added e.g. Figure 3C bottom panel. If the protein is more native, one might expect a more compact distribution of charges (i.e. 4-5 charge states rather than ~13). Also, if multiple conformations are present of the protein contributing to a varied charge state distribution, then why is only one state observed without the standard detergent (e.g. Figure S1E to and bottom spectra comparison)?

The mechanism of charge reduction by polyamines is thought to occur during the nanoESI process where the molecules are surface-active, carrying away protons. The broad charge state distribution is not uncommon when a charge reducing molecule is used. We have made note of this point and note similar observations have been made in previous publications.

Regarding the second point, the ion mobility measurements presented in supplemental show that charge reduction helps retain compact arrival time distributions. These ion mobility measurements (at their current resolution) do not indicate multiple conformations of the protein. The somewhat bimodal charge state distribution (now Figure S2E, bottom panel) is the result of a solution containing mixed charge-reduced molecules.

Reviewer #2 (Remarks to the Author):

The authors here report the synthesis and characterization of four different detergents for native mass spectrometry of membrane proteins. The detergents are composed of polyamine [spermine (SPM)] head group with different hydrophobic alkyl groups: (C4)2SPM, C8SPM, C10SPM and C12SPM. The four derivatives of SPM, along with SPM, are tested for their charge reducing potencies on a variety of

membrane proteins that are expressed in cultured cells and purified. Proteins ranging from monomeric to tetrameric complexes, including TRAAK, AqpZ, β 1AR, AmtB, and TASK1 are tested.

C10SPM and C12SPM are found to be more potent charge-reducing detergents than SPM, C8SPM, and (C4)2SPM. The compatibilities of C10SPM and C12SPM with other commonly used detergents, including decyl maltoside and dodecyl maltoside, are tested and are found to be compatible. In addition, the manuscript also reports weighted average charge states for proteins, TRAAK and AqpZ, in the presence of different ratios of C10SPM to commercial detergents, C10E5 and C8E4, respectively. The mixed micelles display varied charge reduction depending on the ratio of the two detergents, while 100% of C10SPM (1.8 mM) reduces the average charge most (Figure 2). This is mostly true in the case of TRAAK, but the spectrum for AqpZ in 100% C10SPM (Figure S4), however, shows very broad features and makes it difficult for one to see the peaks at lower m/z , thus making it impossible to include those peaks in the calculation of the average charge state. The protein's compact conformers are retained to some extent as per IMS results (Figure S3) though.

From a brief comparison of this study with other studies on charge-reducing additives, listed in the references of this manuscript, it seems that the amount of additive (detergent) required in this study to observe a charge reduction is only 1 mM in most experiments, in comparison to the 40 mM (Imidazole Derivatives, <https://pubs.acs.org/doi/10.1021/acs.analchem.9b04263>), 100 mM (SPM and others, <https://pubs.acs.org/doi/10.1021/acs.analchem.0c01826>) and 100-250 mM (Trimethylamine oxide, <https://pubs.acs.org/doi/10.1007/s13361-019-02187-6>) in previous studies. At 10 mM concentration of various detergents tested here, the spectra of some proteins show broad features (Figures S1, S2, and S4). Notable charge reduction is observed in most cases, except for β 1AR in the presence of the detergent dodecyl maltoside (Figure 4D). The abundance of dissociated species in the protein spectra (Figures 3 and 4) in the presence of decyl maltoside or dodecyl maltoside is reduced due to the addition of C10SPM or C12SPM, owing to their charge reducing properties.

This is not the first time I see this manuscript. The authors have added a significant amount of new data and with that converted the manuscript from a simple proof-of-concept into a truly systematic study. The detergents developed and tested here can simultaneously serve the purpose of membrane protein solubilization and charge reduction. This makes them a valuable addition to the native MS toolbox. I am therefore happy to recommend publication of the manuscript in Nature Communications.

Minor correction: Reference 26 is a duplicate of reference 16.

This has been corrected.